# Review of Automatic Processing of Topography and Surface Feature Identification LiDAR Data Using Machine Learning Techniques

Zahra Gharineiat [ID], Fayez Tarsha Kurdi *[ID] and Glenn Campbell [ID]

School of Surveying and Built Environment, Faculty of Health, Engineering and Sciences, University of Southern Queensland, Springfield Campus, Springfield, QLD 4300, Australia
* Correspondence: fayez.tarshakurdi@usq.edu.au

**Abstract:** Machine Learning (ML) applications on Light Detection And Ranging (LiDAR) data have provided promising results and thus this topic has been widely addressed in the literature during the last few years. This paper reviews the essential and the more recent completed studies in the topography and surface feature identification domain. Four areas, with respect to the suggested approaches, have been analyzed and discussed: the input data, the concepts of point cloud structure for applying ML, the ML techniques used, and the applications of ML on LiDAR data. Then, an overview is provided to underline the advantages and the disadvantages of this research axis. Despite the training data labelling problem, the calculation cost, and the undesirable shortcutting due to data downsampling, most of the proposed methods use supervised ML concepts to classify the downsampled LiDAR data. Furthermore, despite the occasional highly accurate results, in most cases the results still require filtering. In fact, a considerable number of adopted approaches use the same data structure concepts employed in image processing to profit from available informatics tools. Knowing that the LiDAR point clouds represent rich 3D data, more effort is needed to develop specialized processing tools.

**Keywords:** LiDAR; Machine Learning (ML); classification; modelling; point cloud

## 1. Introduction

A Light Detection And Ranging (LiDAR) point cloud (airborne, terrestrial, static or mobile) is a list of 3D points covering the surface of a scanned scene. Topographical data obtained this way are rich in geometric features and lend themselves to the possibilities of automatic processing [1]. There are two major forms of automatic processing operations: automatic classification and automatic modelling [2]. Generally, one scanned scene will consist of classes that have different geometric natures or characteristics, e.g., an urban point cloud represents several classes such as terrain, buildings, vegetation, powerlines, roads, railways, and other artificial objects [3]. As each class in the scanned area will require a different modelling strategy depending on its specific geometric nature, e.g., the vegetation class modelling algorithm will need to be different from the building class modelling algorithm, it is necessary to classify the point cloud before starting the modelling stage.

In the first two decades since LiDAR technology's appearance, most of the suggested automatic processing algorithms belonged to the rule-based family [4]. In truth, a single rule-based algorithm actually consists of a list of procedures connected through a proposed workflow and depends on the physical structure of the point cloud [4]. Recently though, in the domain of topographical LiDAR data processing, the general trend has been to employ Machine Learning (ML) algorithms instead of rule-based ones, and the use of ML techniques has become a popular research topic [5].

Supervised ML algorithms assign observations to data classes previously generated, either manually or automatically, from the use of training data that could sometimes be

generated automatically [6]. Alternatively, unsupervised ML algorithms do not need training data and can be classified into four families: classification tree methods such as the Random Forest (RF) algorithms, grouping and separability methods such as Support Vector Machines (SVM), k-Nearest Neighbors (KNN), and rule application methods such as Convolutional Neural Networks (CNN) [7].

This paper reviews the state-of-the-art ML algorithms developed for topographical LiDAR data processing. The novelty of this paper is the classification and analysis of the ML algorithms according to four different dimensions. First, the methods of point cloud generation for input into ML approaches are analyzed and discussed. Second, the different concepts of point cloud structure that are commonly used are studied and compared. Third, the suggested approaches are classified according to the most employed ML techniques, and then the main ML techniques are summarized. Finally, the most current applications of ML techniques are classified and cited.

## 2. Input Data

Notwithstanding the quality of the employed laser scanning technology, airborne, terrestrial, static, or mobile, all methods allow the creation of a 3D point cloud that covers the scanned area. A LiDAR point cloud consists of a point list of co-ordinates *X*, *Y*, and *Z* defined in 3D Euclidean space. For each point, in addition to the three coordinates, laser intensity, waveform, and Red Green Blue (RGB) colors can be provided [8]. Furthermore, for the same scanned scene, additional data such as multispectral images, maps, and orthophotos can often be provided. As a result, in the literature, the suggested ML approaches for LiDAR data processing are not just limited to the LiDAR point cloud alone. The following subsections explain the different point cloud generation methods for input into ML algorithms.

### 2.1. LiDAR Point Clouds

The 3D point cloud is the primary output of a laser scanning operation (Figure 1). This subsection deals with approaches that use only the point cloud, whereas the approaches that use other additionally acquired data will be discussed in the following subsections. The obvious advantage of approaches that use only the LiDAR point cloud is that they are always available for use in all scanning projects. The point cloud does not just represent a simple list of 3D points in the Euclidian space, it may be used as the input data to create a Digital Surface Model (DSM) [1]. Furthermore, for each point, a list of neighboring points can be defined in 3D space [9–11], where all points included inside a sphere surrounding the focus point are considered, or in 2D space where all points included inside a cylinder surrounding the focus point are considered [5]. After this stage is completed, each point and its neighboring points allow for fitting a mean line or plane to analyze their relative topologic positions through several indicators such as standard deviation, mean square error, eigenvector, and eigenvalues [12]. Additionally, the eigenvector permits the calculation of a list of useful geometric features such as linearity, planarity, sphericity and change of curvature [13,14]. In this context, other approaches are to superimpose the point cloud on an empty 2D grid to allow for the analysis of the topological relationships between neighboring points [15], or assuming that they represent one object, using one LiDAR point and its neighborhood to allow calculation of a list of static moments that help to study some of their geometric characteristics [16]. While this has its uses, it is important to note that the employment of just the point cloud as input data does not produce promising results in the general case, e.g., when identifying roofs in an airborne LiDAR point cloud in an urban area, the range of roof point coordinates may be incorrectly allocated to an incorrect building because of underlying topography of the scanned area. That is why the application of the ML techniques in this case uses the point features instead of the point coordinates as input data [9]. Consequently, a long list of geometric features that can be calculated from the point cloud is needed to create a suitable environment to apply ML. The ML techniques that have been applied to airborne and terrestrial LiDAR point clouds are

shown in the next two subsections, with the use of laser intensity observations discussed in the subsection that follows those.

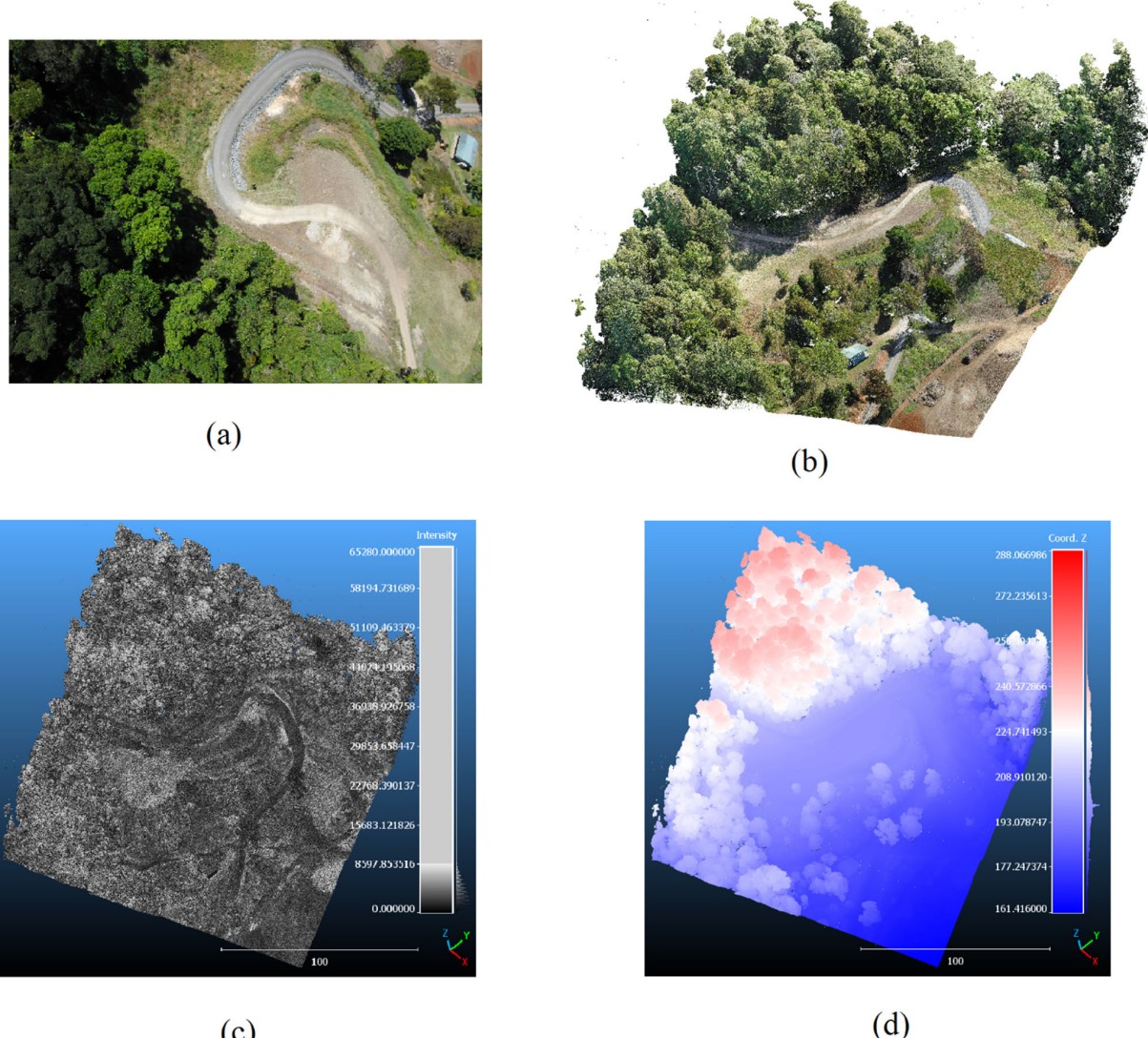

**Figure 1.** (**a**) Aerial image of scanned scene; (**b–d**) 3D LiDAR point cloud visualization (**b**) using RGB colors; (**c**) using laser intensity values; (**d**) using Z coordinate values.

### 2.1.1. Airborne LiDAR Point Cloud

Airborne LiDAR point clouds provide two obstacles to the applications of ML techniques: variation in point density within the scanned scene [11] and the large number of LiDAR points [17]. Point density plays a vital role in selecting the neighboring points for the calculation of point features [9]. Point density can vary markedly within the same point cloud with the location within the scanning strip, the terrain topography and the geometry, and the orientation of the scanned object with regard to the scan line all having an affect [8]. For a large area, the data volumes can be excessive, meaning the training step will place heavy demands on the computer capacity and processing time [17,18]. Lin et al. [19] and Mao et al. [20] developed approaches to mitigate this problem and classify an urban point cloud into nine classes: powerlines, low vegetation, impervious surfaces, cars, fences, roofs, façades, shrubs, and trees. In this context, Mao et al. [20] developed a Receptive Field Fusion-and-Stratification Network (RFFS-Net). An innovative Dilated Graph Convolution (DGConv) and its extension, the Annular Dilated Convolution (ADConv), are fundamental components of elementary building blocks. The receptive field fusion procedure was applied with the Dilated and Annular Graph Fusion (DAGFusion) component. Thus, the

detection of dilated and annular graphs with numerous receptive zones allows the acquisition of developed multi-receptive field feature implementation to improve classification accuracy. To efficiently extract only one class from the urban point cloud, Ao et al. [21] advised using a presence and background learning algorithm like a backpropagation neural network.

### 2.1.2. Terrestrial LiDAR Point Cloud

This subsection focuses only on the ML approaches that use a static or mobile terrestrial LiDAR point cloud as input data either indoors or outdoors. An indoor cloud may focus on certain scanned objects such as tables, chairs, decorative statues, and mechanical equipment [22,23] or it may carry out a panoramic scan [24] and use the LiDAR point cloud to then extract the individual objects. An urban outdoor LiDAR point cloud will most likely emphasize artificial or natural objects such as building facades and terrain [25] while a rural scene, like Zou et al. [26] examined, may use a terrestrial LiDAR point cloud of forestry areas to classify the tree species. In fact, most of the suggested approaches that use ML techniques to process terrestrial LiDAR data do not use additional data with the point cloud [17,27–32]. Point density variation has less influence in terrestrial when compared to airborne data. Nevertheless, some authors do use additional data as input, e.g., Xiu et al. [23] suggested a ML algorithm to process indoor point cloud represented by 9 dimensions: *X, Y, Z, R, G, B*, and normalized location. He et al. [25] developed a SectorGSnet framework for a ground segmentation of terrestrial outdoor LiDAR point clouds. This framework consisted of an encoder in addition to segmentation modules. It introduced a bird's-eye-view segmentation strategy that discretizes the point cloud into segments of different areas. The points within each partition are then fed into a multimodal Point-Net encoder to extract the required features. Li et al. [33] suggested a Rotation Invariant neural Network (RINet) which associated semantic and geometric features to improve the descriptive capacity of scanned objects and classify the terrestrial data into twelve classes.

Terrestrial laser scanning plays a major role in autonomous driving vehicles with Silva et al. [34] developing a Deep Feature Transformation Network (DFT-Net) involving a cascading mixture of edge convolutions and feature transformation layers to capture the local geometric features by conserving topological relationships between points. Alternatively, self-learning algorithms appear as a practical solution to understand the correspondence between adjacent LiDAR scan scenes [35]. Nunes et al. [36] used a momentum encoder network and a feature bank in a self-learning approach [37,38] that aimed to learn the structural context of the scanned scene. This approach applies the contrastive loss over the extracted segments to distinguish between similar and dissimilar objects. Finally, Huang et al. [39] used an unsupervised domain adaptation ML to classify terrestrial LiDAR data and suggested using Generative Adversarial Network (GAN) to calculate synthetic data from the source domain, so the output will be close to the target domain.

### 2.1.3. Point Cloud and Laser Intensity

In practice, LiDAR systems measure and provide the laser pulse return intensity (Figure 1c). The intensity of emitted laser pulse is greater than the intensity of the reflected laser pulse and with the difference being dependent on the double distance trajectory in addition to the nature of the reflecting surface off which the pulse has returned [40]. Unlike the RGB-measured values of the point cloud, the intensity could be detected regardless of the illumination and can be provided in both airborne and terrestrial LiDAR. Some authors have used the intensity and the 3D point cloud together as input data into their ML algorithms.

In this regard, Wen et al. [41] proposed a Directionally constrained fully Convolutional Neural network (D-FCN) where the input data were the original 3D point cloud in addition to the LiDAR intensity. Since road line markings have a higher reflectance, and hence higher intensity value than the surrounding ground, Fang et al. [42] considered the 3D LiDAR

point cloud and the laser intensity as input data to their ML algorithm. Wang et al. [29] employed the intensity component in semantic outdoor 3D terrestrial dataset to achieve the cloud segmentation using Graph Attention Convolution (GAC) and Murray et al. [43] calculated a 2D image from the intensity component of LiDAR data. This image was used as input data for the CNN algorithm and then for the SVM.

### 2.2. Point Cloud and Imagery

In the image processing domain, many algorithms for feature extraction from images have been implemented where the image's spatial and textural features were extracted using mathematical descriptors, such as histograms of oriented gradients and SVMs [44]. The combination of LiDAR data with high-resolution images can provide highly relevant data for the analysis of scanned scene characteristics [45]. Indeed, numerous authors develop classification ML networks using LiDAR point clouds in addition to digital images as input data. Nahhas et al. [46] employed orthophotos in addition to airborne LiDAR point clouds to recognize the building class by using an autoencoder-based dimensionality reduction to convert low-level features into compressed features. Similarly, Vayghan et al. [3] used aerial images and LiDAR data to extract building and tree footprints in urban areas while Zhang et al. [47] fused the LiDAR data and a point cloud calculated from the aerial images to improve the accuracy of a ML building extraction algorithm. Shi et al. [48] suggested the use of an enhanced lightweight deep neural network with knowledge refinement to detect local features from LiDAR data and imagery while preserving solid robustness for day-night visual localization.

### 2.3. Multispectral LiDAR Data

Multispectral images have layers that represent the reflectance in a few wide and disconnected spectral bands within given specified spectral intervals [49]. In the case of airborne LIDAR data, some authors have used multispectral images in addition to the LiDAR point cloud as input data for ML algorithms, because most objects on the Earth's surface have indicative absorption features in certain discrete spectral bands which can help to create an accurate classification of the scanned scene [49]. Though the multispectral data are not always available, where they are, they can be an asset for processing efficacy. In this context, Marrs and Ni-Meister, [50] used LiDAR, hyperspectral, and thermal images on experimental forests and found that the combination of these two data can help improve the classification of tree species. Yu et al. [51] used multispectral LiDAR data for individual tree extraction and tree species recognition. Zhao et al. [52] used a FR-GCNet network to increase the classification accuracy of multispectral LiDAR point clouds, whereas Zhou et al. [53] applied an RF algorithm on a combination of hyperspectral images and LiDAR data for monitoring insects. Peng et al. [54] suggested a MultiView Hierarchical Network (MVHN) could be used to segment hyper spectral images and LiDAR point cloud together. For this purpose, the hyper spectral images were divided into multiple groups with the same number of bands to extract spectral features. Thereafter, ResNet framework was implanted to detect the spectral-spatial information of the merged features.

### 2.4. Full-Waveform Representation and Point Cloud

Some airborne laser systems, called full waveform, can record the complete power spectrum of the returned pulse. The different surface characteristics can influence the reflected signal, so analysis of the laser pulse full waveform has been used to improve the extraction of surface features [55] especially in forested areas. Five parameters are calculated from the waveform of the return pulse, e.g., the amplitude of the highest peak, the total energy, the full-width half-maximum return width, and the length of the sequence. In this context, Guan et al. [56] constructed a geometric tree model based on the full-waveform representation. Afterward, in order to classify the tree species, they applied a deep learning algorithm to the last model to extract the high-level features. Blomley et al. [57] classified

the tree species using the RF algorithm based on the geometric features calculated from the full-waveform analysis.

Similarly, by means of an integrated system that acquired hyperspectral images, LiDAR waveforms, and point clouds, Yang et al. [58] classified tree species after systematic pixel-wised investigation of different features. For this purpose, the Canopy Height Model (CHM) was extracted from the LiDAR data, and multiple features from the hyperspectral images, including Gabor textural features. Shinohara et al. [59] suggested a semantic classification algorithm named Full-Waveform Network (FWNet) based on PointNet-based architecture [27], which extracted the local and global features of the input waveform data. The classifier in this case consisted of 1D convolutional operational layers. Due to the sensitivity of border points to the multi return difference value, to achieve the cloud segmentation, Shin et al. [60] used multiple returns in addition to the point cloud as training data using the PointNet++ network [61].

*2.5. Different Other Data*

Sometimes other data, not mentioned previously, may be used in addition to the LiDAR point cloud. For example, Zhang et al. [62] used the interaction of the high-resolution L-band repeat-pass Polarimetric Synthetic Aperture Radar Interferometry (PolInSAR) and low-resolution large-footprint full-waveform LiDAR data to estimate forest height. Park and Guldmann [63] utilized a city LiDAR point cloud in addition to building footprint data to extract building class before applying an RF algorithm and Feng and Guo [64] suggested a segment-based parameter learning approach that fuses a 2D land map and 3D point cloud together.

For detecting individual trees, Schmohl et al. [65] used an orthophoto to colorize the point cloud for additional spectral features along with laser intensity and the number of returns utilized as additional input. Kogut et al. [66] improved the classification accuracy of seabed laser scanning (bathymetry data) by using the Synthetic Minority Oversampling Technique (SMOTE) algorithm to evaluate the input data. Then, a Multi-Layer Perceptron (MLP) neural workflow was applied to classify the point cloud. Barbarella et al. [67] applied a ML network that trained a model able to classify a particular gravity-driven coastal hillslope geomorphic model (slope-over-wall) including most of the soft rocks. However, they used only geometric data which are morphometric feature maps computed from a Digital Terrain Model (DTM) calculated from the LiDAR point cloud.

Finally, Duran et al. [68] compared nine ML methods: logistic regression, linear discriminant analysis, K-NN, decision tree, Gaussian Naïve Bayes, MLP, adaboost, RF, and SVM to classify LiDAR and colored photogrammetric point clouds into four classes: buildings, ground, low and high vegetation with the highest accuracy being attained with MPL. For more details about these ML techniques, please see Mohammed et al. [69] and Kim, [70].

## 3. Concepts of Point Cloud Structure for Applying ML Algorithms

The 3D point cloud consists of a large number of 3D points covering the scanned area. These points are normally distributed in an irregular way depending on the scanning system quality and the scanned area geometric characteristics. In any event, to process, classify, and model the LiDAR data using ML techniques, most of the suggested approaches try to define a mathematical model that allows for the management, reduction, pooling, and convolution of these data [71]. Consequently, most ML approaches consist of two main steps, firstly preprocessing and then ML algorithm application. In this paper, the mathematical model in addition to all operations realized on it before applying the ML technique is named the data adaptation step (Figure 2). The data adaptation procedures may play several roles. Some ML informatics tools for imagery data processing or other data kinds, require the transformation of point cloud into novel data forms such as 2D and 3D matrices before they can be used. As informatics tools for processing LiDAR data require high time processing cost, two solutions are employed: either designing new ML tools that correspond to the LiDAR data concept or, more commonly, reducing the LiDAR

data. At this stage, it is important to refer that the interpolation or reduction of LiDAR data is not always a preferable solution from the geomatics industrial viewpoint.

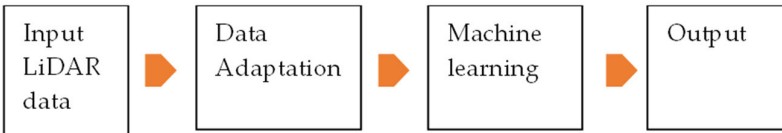

**Figure 2.** Structure of ML algorithm of LiDAR data processing.

In the next subsections, the main concepts of LiDAR data adaptation will be revealed and discussed.

### 3.1. Voxelization

Voxelization, a 3D matrixial representation, may sometimes solve the issue of the irregular distribution of the 3D point cloud [56]. In practice, the LiDAR points are distributed on the scanned surfaces which leads to a considerable number of empty voxels which cause additional calculation costs. Moreover, course spatial resolution (large voxel size) may cause the loss of information which will reduce the accuracy of data processing. Conversely, if the spatial resolution is too small, that may increase the calculation cost, and the memory usage [17].

In the literature, many authors suggest voxelizing the LiDAR point clouds. In this context, Maturana and Scherer [72] developed the VoxNet network using the occupancy grid algorithm. They divided the point cloud into many 3D grids and then normalized each grid unit to enter the volume build layers and maximum pooling layers. Gargoum et al. [73] suggested a voxel-based approach to classify the light poles of roads while Zou et al. [26] proposed a voxel-based deep learning method to identify tree species in a three-dimensional map. They extracted individual trees through point cloud density and used voxel rasterization to obtain features. Guan et al. [56] used a voxel-based upward growth algorithm to remove the ground point cloud and then segment a single tree species by European clustering and a voxel-based normalization algorithm. Shuang et al. [74] developed an Adaptive Feature Enhanced Convolutional Neural Network (AFERCNN) for 3D object detection. This algorithm is a point-voxel integrated network, where voxel features are extracted through the 3D voxel convolutional neural network. These features are projected to the 2D bird's eye view and the relationship between the features in both spatial dimension and channel dimension is learned. Wijaya et al. [75] applied a voxel-based 3D object detection deep neural network on terrestrial LiDAR data where they minimized the features from a 3D into a 2D bird-eye view map before generating object proposals to save processing time.

However, voxelization tries to conserve the LiDAR point cloud 3D structure by defining a spatial matrixial form that enables improved management of the point cloud. Hence, the form will be limited by the available, the used memory, and the requested processing time may represent the main limitations.

### 3.2. Graphic Structure

Using graphic structure to transform the 3D point cloud into a 2D regular grid has the main advantage of transforming the point cloud classification question into the general image processing one. Simonovsky and Komodakis [76] used edge labels to calculate Edge Conditional Convolution (ECC) in the neighborhood of regular grids. Then, an asymmetric edge operation was used to calculate the relationship between neighboring points. Wang et al. [77] developed a SpecGCN network where the maximum pooling was replaced with a recursive clustering. The nearest neighbor was applied to calculate a graph regular grid. Thereafter, they combined a spectral graph convolution using a local graph, with a pooling strategy. Nahhas et al. [46] suggested a deep learning approach based on using an interpolated LiDAR point cloud and orthophotos simultaneously. This approach employed object-based analysis to create objects, a feature-level fusion. Li et al. [78] devel-

oped a deep learning network named Attentive Graph Geometric Moments Convolution (AGGM Convolution) network to classify the LiDAR point cloud into four classes: trees, grass, roads, and buildings. The Dynamic Graph Convolution Neurol Network (DGCNN), suggested by Wang et al. [28], built the directed graph in both the Euclidean space and the feature space, and dynamically updated the feature layers. A similar approach suggested by Wang et al. [29] employed the attention mechanism in the graph-based methods. The extended approach is named Graph Attention Convolution Network (GACNet) for semantic point cloud segmentation.

In the same context, Wen et al. [79] presented a global-local Graph Attention Convolution Neural Network (GACNN) that could be directly applied to airborne LiDAR data. The graph attention convolution module includes two types of attention mechanisms: a local attention module that combines edge attention and density attention, and a global attention module. The local edge attention module is designed to dynamically learn convolution weights using the spatial relationships of neighboring points; thus, the receptive field of the convolution kernel can dynamically adjust to the structure of the point cloud. Zhao et al. [52] used a Feature Reasoning-based Graph Convolution Network (FR-GCNet) to increase the classification accuracy of airborne multispectral LiDAR data. Jing et al. [80] proposed a Graph-based neural Network with an Attention pooling strategy (AGNet) where the local features were extracted through the point topological structure. Chen et al. [81] improved the descriptiveness in the network ChebyNet [82] by increasing the width of input to avert the above drawbacks. The suggested network, named WGNet, is inspired by the image processing dilated convolution. This network is based on two modules, the local dilated connecting and context information awareness. Wan et al. [83] developed a Dilated Graph Attention-based Network (DGANet) for local feature extraction on 3D point clouds. It was based on the dilated graph attention modules which allow the network to learn the neighborhood representation by using the long-range dependencies given by the calculated dilated graph-like region for each point.

To conclude, the use of graphic structure facilitates the point cloud processing duty tasks by using image processing functions, but unfortunately at the cost of minimizing the 3D structure advantages.

### 3.3. Kernel-Based Convolution

The geometric structure of a point cloud can be defined through the Kernel correlation layer [41]. The kernel size value can be suggested according to a different number of neighboring points in the convolution layer. Points within the kernel can contribute to their center point [84]. At this stage, Klokov et al. [85] proposed a K-NN algorithm that uses the Euclidean metric to return the closest points inside the kernel. The kernel is defined by two parameters: the inner and the outer radius to ensure that the closest and unique points will be detected in each ring kernel. In the context of ML applications, Song et al. [86] employed the kernel correlation learning block approach to recognize the local and global features at different layers thus enhancing the network perception capacity. Zhang et al. [31] suggested a Local k-NNs Pattern in Omni-Direction Graph Convolution Neural Network named LKPO-GNN to capture both the global and local point cloud spatial layout. This approach converts the point cloud into an ordered 1D sequence, to feed the input data into a neural network and reduce the processing cost.

In fact, this approach allows applying all operations directly on the point cloud, but it still requires an optimized neighborhood searching procedure.

### 3.4. Reducing of Point Cloud Density (Downsampling)

Most ML approaches applied to LiDAR data try to reduce data density and keep the processing time within accepted limits. The successful use of the convolutional technique within the image processing field has encouraged authors to use the same approach in reducing LiDAR data and thus to solve the processing time issue. Although the most used point cloud structures apply the idea of point cloud reduction, the suggested approaches in

this subsection conserve the point cloud structure and reduce the point density. However, the application of ML techniques is still in its infancy, and a lot of advancement is expected in future research.

In the context of point cloud reduction, Wen et al. [41] developed a D-FCN network architecture that included both downsampling and upsampling paths to enable multiscale point feature learning. Several authors, Hu et al. [30], Wei et al. [17], and Du et al. [22] used random downsampling to reduce the point cloud in the context of applying the ML algorithm such as developing consecutively feed-forward MLPsRandLA-Net and encoder–decoder structure (BushNet and ResDLPS-Net). Mao et al. [20] suggested three downsampling layers to classify the LiDAR data.

Though the downsampling reduces the data volume, it loses an important information quantity that may be useful to object recognition and modeling.

## 4. Employed ML Techniques

Currently, the advancement of digital technologies and data acquisition techniques in different disciplines can lead to the generation of excessively large data sets. To manage and process the oversized data sets, the questions of data classification and object recognition have become ones of crucial importance. In this context, ML techniques occupy an enviable position because they allow for automatic and efficient solutions. The ML techniques can be classified into four categories according to the required input data (see Mohammed et al. [69]): supervised learning, where labelled data are needed for training, unsupervised learning, where labelled data are not needed, semi-supervised learning that uses a mixture of classified and unclassified data, and reinforcement learning where no data are available. Of these, the supervised and unsupervised techniques may be considered the main two categories. In each one of these two groups, several algorithms are employed, e.g., supervised ML uses algorithms such as decision trees, rule-based classifiers, Naïve Bayesian classification, k-nearest neighbors' classifiers, RF, Neural Networks (NN), linear discriminant analysis, and SVM, whereas unsupervised ML uses k-means clustering, Gaussian mixture model, hidden Markov model, and principal component analysis.

In the LiDAR data-processing domain, the application of ML algorithms represents an emerging research area. Despite the great number of papers published in this area, very few new ML algorithms are employed. In the next subsections, more focused ML algorithms will be introduced and discussed.

### 4.1. Random Forest (RF) and Support Vector Machine (SVM)

Tarsha Kurdi et al. [87] summarized the applications of RF classifiers for automatic vegetation detection and modelling using LiDAR point clouds. Many authors used RF exclusively on LiDAR data [88], whereas other authors used additional data [89,90]. Yu et al. [91], and Yu et al. [51] estimated tree characteristics such as diameter, height, and stem volume using an RF classifier and Levick et al. [92] connected the DSM and field-measured wood volume using an RF algorithm. Chen et al. [88] used the feature selection method and an RF algorithm for landslide detection under forest canopy, where the DTM and the slope model were constructed for the scanned area, and the features were calculated at the pixel level. The same principle was used by Guan et al. [93] to identify the city classes in urban areas and Ba et al. [94] employed RF for detecting the tree species.

Man et al. [90] applied an RF classifier to calculate a two-dimensional distribution map of urban vegetation. In this study, individual tree segmentation was conducted on a CHM and point cloud data separately to obtain three-dimensional characteristics of urban trees. The results show that both the RF classification and object-based classification could extract urban vegetation accurately, with accuracies above 99%, and the individual tree segmentation based on point cloud data could delineate individual trees in three-dimensional space better than CHM segmentation. Arumäe et al. [95] calculated a model for predicting necessity thinning using the RF technique to retrieve the two indicative parameters for requiring thinning, height percentage and the canopy cover. Park and

Guldmann, [63] used an RF algorithm to classify building point clouds into four classes: rooftop, wall, ground, and high outlier. To overcome the complexity of building geometry of the Ming and Qing Dynasties' Official Architecture style (MQDOAs), Dong et al. [96] employed semantic roof segmentation. This method was composed of two stages. Some geometric features such as the normalized symmetrical distance, relative height, and local height difference are extracted and then the RF algorithm is applied to classify the roof point cloud. Feng and Guo [64] suggested a segment-based parameter learning approach in which a 2D land cover map is chosen to generate labelled samples, and a formalized operation is then implemented to train the RF classifier. Liao et al. [97] fed in point cloud super voxels and their convex connected patches into an RF algorithm. For this purpose, they consider three types of features: point-based, eigen-based, and grid-based.

The SVM algorithm tries to find a hyperplane in high dimensional feature space to classify some linearly correlative point distributions. While there could be many hyperplanes that separate the target classes, the hyperplane that optimizes the boundary between the classes is identified. Aside from just linear classification, SVM can carry out nonlinear classification using the kernel trick by indirectly drawing their inputs into high-dimensional feature spaces [69].

Though the SVM classifier is efficient for data classification when using rather small data, it is also used by Ba et al. [94] to recognize tree species. Murray et al. [43] trained an SVM on the passing and ongoing results of a CNN algorithm through pixel classification and the interpolation result of the intensity vector as input data. Hoang et al. [98] introduced a hybrid approach of a CNN and an SVM for 3D shape recognition, where eight layers of the CNN are utilized for geometric feature extraction and afterward an SVM is applied to classify them. Zhang et al. [99] suggested an object-based approach to classify an urban airborne LiDAR point cloud. First, different point features such as geometry, radiometry, topology, and echo characteristics are extracted and then the SVM classifier algorithm was applied to detect five classes: terrain, vegetation, building, powerlines, and vehicles. To detect powerlines, Shokri et al. [100] eliminated the undesirable points and then apply the SVM after calculating the point geometric features.

In conclusion, RF and SVM are less used in recent years, and both are more basic classification models. Therefore, most modern approaches focus on deep learning techniques.

*4.2. Neural Network and Deep Learning*

Deep learning represents a sort of ML, and it can be defined as a ML technique that employs a deep neural network such as the MLP neural network that contains two or more hidden layers [70]. A Perceptron Neural network consists of single neurons that have multiple inputs and generate a single output using an activation function. Figure 3 illustrates a deep learning algorithm functionality where the available data consist of two sections: labelled and unlabeled data. The labelled data will be used in training the suggested MLP neural network to correct the assumed weight values which will then be used in the same neural network to label the unlabeled data. For more information about deep learning techniques, please see Kim [70].

In the LiDAR data processing area, deep learning algorithms are widely applied especially for data classification. Zou et al. [26] used a low-level feature representation through voxel-based structure, and then classified tree species by using a deep learning model. In regard to Generative Adversarial Networks (GAN), Goodfellow et al. [101] have achieved a notable performance on pan-sharpening in the image processing domain. Zhang et al. [62] developed a PolGAN deep learning network to determine the forest tree heights. When applying a deep learning classification algorithm, Lin et al. [19] improved the labelling stage to produce training data because the data labelling procedure for generating training data consumes considerable time and effort. In this context, they suggested using weak labelling that needs little annotation effort. The pseudo labels are then considered as the input of a classification network [102]. Thereafter, an overlap region loss and an

elevation attention unit are introduced for the classification network to obtain more accurate pseudo labels.

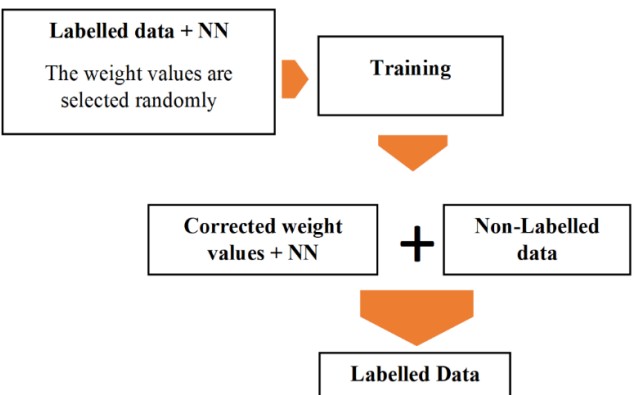

**Figure 3.** Deep learning functionality; NN is a Neural network.

Zhao et al. [52] used a Feature Reasoning-based Graph Convolution Network (FR-GCNet) to increase the classification accuracy of urban point clouds. Semantic labels were assigned to pixels using global and local features. Based on the graph convolution network, a global reasoning unit is embedded to find the global contextual features, while a local reasoning unit is added to learn edge features with attention weights in each local graph. Li et al. [103] compared three deep learning algorithms for classifying LiDAR point clouds, these algorithms are PointNet++ [61], SparseCNN [104] and KPConv [105]. They found that SparseCNN carries out a better classification accuracy than the other two approaches.

Where there are variations of point cloud density, Théodose et al. [106] suggested adapting an object detection deep learning approach. For this purpose, some data layers are randomly dismissed during the training step to grow the variability of the processed data. Sheikh et al. [32] proposed a Deep Feature Transformation Network (DFT-Net) to classify terrestrial LiDAR data. The suggested algorithm is based on graph analysis in which the edges are dynamically extracted for each layer. Hoang et al. [107] extracted and associated both global and regional features through Gaussian SuperVector and enhancing region illustration deep learning Network (GSV-NET) for 3D point cloud classification. Chen et al. [108] developed a Dynamic Point Feature Aggregation deep learning Network (DPFA-Net) by selectively performing the neighborhood feature aggregation, dynamic pooling, and an attention mechanism. In this semantic classification of the LiDAR point cloud framework, the features of the dynamic point neighborhood are aggregated via a self-attention mechanism. Finally, Song et al. [109] developed, in the context of automatic LiDAR data classification, a 2D and 3D Hough Network (2D&3DHNet) by linking 3D global Hough features and 2D local Hough features with a classification deep learning network.

### 4.3. Encoder–Decoder Structure

In the encoder–decoder structure, the network consists mainly of two subnetworks: the encoder sub-network and the decoder sub-network [110]. In the encoder part, consecutive downsampling procedures increase the receptivity of the extracted features but unfortunately, that reduces the point cloud resolution. In the decoder part, upsampling and convolution operations are employed for resolution recapture and feature combination.

In laser scanning, several authors developed an encoder–decoder algorithms to classify LiDAR data. Wen et al. [79] created an end-to-end encoder–decoder network named GACNN that is based on the graph attention convolution module and used it for detecting multiscale features of the LiDAR data and achieving point cloud classification. Wei et al. [17] proposed a network point cloud segmentation named BushNet which is the classic encoder–decoder structure. In this context, a minimum probability random sampling module is used for reducing the processing time and improving the convergence speed. Thereafter, the local multi-dimensional feature fusion module is applied to make the

network more sensitive to bush point cloud features. Thus, the employed multi-channel attention module may improve the training efficiency.

Medina and Paffenroth [111] applied an encoder–decoder classifier for reduced LiDAR data by feeding the network with features calculated from the point neighborhood, which showed high efficiency in distinguishing the non-linear features. Mao et al. [20] developed an encoder–decoder architecture for point cloud classification named Receptive Field Fusion and Stratification Network (RFFS-Net) that is based on the PointConv network suggested by Wu et al. [112]. It consists of two steps: hierarchical graph generation and encoder–decoders feature extraction and aggregation. The input is provided by a hierarchical graph generation model and point features after which the point features are aggregated. Ibrahim et al. [113] used CNN architectures to semantically classify the terrestrial LiDAR data. They divided the point cloud into angle-wise slices that are transformed in the next step into enhanced pseudo images using the intensity and reflectivity values. Then, these images are employed to feed an encoder–decoder CNN model.

Finally, despite the promising results obtained by deep learning as well as encoder–decoder structure, more focus is needed on unsupervised learning techniques which may cancel the request for training data.

Having presented the main ML algorithms used to process LiDAR data, the next section will discuss current applications of ML technique on LiDAR point cloud.

## 5. Applications of ML on LiDAR Data

The use of laser scanning technology is widespread. It has been applied in urban, rural, and forested areas to target natural as well as artificial objects such as buildings (inside and outside), roads, railways, bridges, tunnels, and pipelines. Almost inevitably, a point cloud of a scanned area will consist of several object classes such as terrain, vegetation, buildings, standing water, noise, and artificial objects. As each class has a different modelling concept, it is essential to classify the point cloud into its main classes before starting the modelling step [5]. Once the point cloud of the scanned area is classified, the obtained classes can be analyzed and modelled according to the project goal. In this context, a large list of class modelling operations could be described. From the creation of laser scanning technology, most of the suggested approaches in the literature have been rule-based. Within the last five years, ML techniques have become an important approach for LiDAR data processing [2,8]. Unfortunately, ML techniques have hitherto only been used to a limited number of procedures, e.g., according to Hamedianfar et al. [114], the main applications of deep learning algorithms in forest areas are biomass estimation and tree species classification. In the next subsections, the main applications of ML techniques on LiDAR data are detailed.

### 5.1. Building Detection

The ML classifiers are sometimes focused on the building class in urban areas, with the aim of classifying the scanned scene into two classes: buildings and non-buildings. Nahhas et al. [46] suggested a deep learning approach based on the feature-level. CNN was used to transform compressed features into high-level features, which were used in building detection. Zhang et al. [47] used the U-NET model [115] to detect building polygons from orthophotos. Hence, to increase the point cloud density, the LiDAR, and photogrammetric point clouds are merged and employed for each polygon for feature extraction goals. Ojogbane et al. [116] improved a deep learning network suggested by Seydi et al. [117] to detect the building class. The suggested framework fuses the features obtained from interpolated airborne LiDAR data into DSM, in addition to a very high-resolution aerial imagery. Shin et al. [60] applied PointNet++ [61] for building extraction using multiple returns data.

While ML algorithms are employed by several authors for building recognition, in fact, the urban scene cannot just be simplified into building and non-building classes. Hence, the next section will go further through applying ML to achieve full classification.

### 5.2. Scene Segmentation

The classification question is widely discussed in this research area. One scanned scene consists of several classes, and the question that arises is: can the classification algorithm be used to extract the desired class list? Or can one algorithm only recognize certain classes? For this reason, we have chosen to identify the classification algorithms according to detected classes. With respect to airborne data, not all authors agree about the ideal number of classes. Wen et al. [41] developed a deep learning network that classified the airborne LiDAR data into nine classes: powerlines, low vegetation, cars, fences, roofs, facades, shrubs, and trees. Despite Wang and Gu [118] using the same number of classes, their class list is different: earth bar, grass, roads, buildings, trees, water, powerlines, cars, and ships. Li et al. [78] suggested a deep learning pixel-based analysis network to distinguish four classes in airborne data: trees, grass, roads, and buildings. Another class list is suggested by Ekhtari et al. [119] classified their scene into six classes: buildings, soil, grass, trees, asphalt, and concrete. An example of the final data set is shown in Figure 4. Zhao et al. [52] made small modifications to these classes as follows: roads, buildings, grass, trees, powerlines, and soil. Another modification to these classes is suggested by Shinohara et al. [59]: roads, buildings, transmission towers, trees, powerlines, and ground. Liao et al. [97] classified the airborne point cloud into three main classes: terrain, buildings, and vegetation using the RF algorithm. Zhao et al. [120] suggested a Point Expanded Multi-Scale Convolutional Network (PEMCNet) to classify the airborne LiDAR data containing point cloud, intensity, and return number, into five classes: ground, high vegetation, building, water, and raised road. To calculate the point features, it created point expanded grouping units that combined the extracted features at diverse scales. It is fair to say that the classes chosen in each study are a product of the study area and study aim rather than a desire to develop a universal class set.

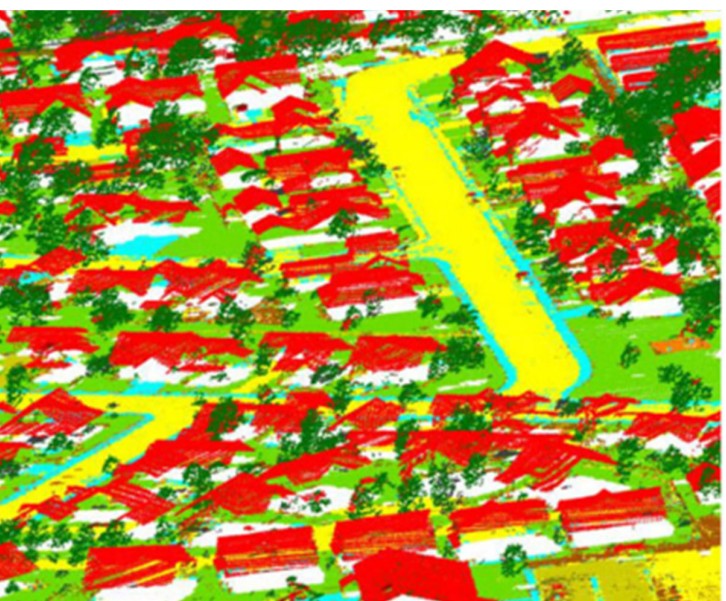

**Figure 4.** An example of a 3D point cloud classified into six classes (buildings, soil, grass, trees, asphalt, and concrete) by Ekhtari et al. [119].

In the case of terrestrial data, a huge diversity of suggested class lists reflects the diversity of scanned scenes. Wang et al. [28], Qi et al. [27], Wang et al. [29], Hu et al. [30], Wei et al. [17], Zhang et al. [31], Xiu et al. [23], and Jing et al. [80] classified the terrestrial LiDAR data into several classes according to the scanned objects. To classify the terrestrial LiDAR data, Wen et al. [121] converted the LiDAR point cloud into a pseudo image and applied a semantic segmentation algorithm named Hybrid CNN-LSTM that has a neural network framework. Hence, the pseudo image is considered within Long Short-Term

Memory (LSTM) network that combines the different channel features generated by a convolutional neural network. Shuang et al. [122] proposed for terrestrial LiDAR point cloud classification, a Multi-Spatial Information and Dual Adaptive (MSIDA) network, which consists of encoding and dual adaptive sub-networks. To encode the point coordinates, each point and its neighborhood are transferred into a cylindrical and spherical coordinate system. The DA sub-network comprises a Coordinate System Attention Pooling Fusion (CSAPF) block in addition to a Local Aggregated Feature Attention (LAFA) one.

### 5.3. Vegetation Detection

Some classification algorithms are developed especially for forest areas, that focus on the vegetation class. In this case, they classify the scanned scene into two classes: vegetation and non-vegetation. Luo et al. [24] developed a semantic segmentation deep network to extract vegetation points from the LiDAR point cloud, where the tree points are grouped into a set of tree clusters using Euclidean distance clustering. A Pointwise Direction Embedding deep network (PDE-net) is employed to calculate the direction vectors of tree centers. Chen et al. [123] compared four ML algorithms: RF, Cubist, XGBoost, and CatBoost with rule-based algorithms to improve the estimation performance of forest biomass. The ML algorithms outperformed parametric stepwise regression, with the CatBoost network being superior, followed by XGBoost, RF, Cubist, and stepwise regression.

In the context of individual tree detection, Schmohl et al. [65] exploited the 3D LiDAR point cloud by using a 3D NN to detect individual trees. A sparse convolutional network was applied for feature calculation and feeding of the semantic segmentation output. Furthermore, they defined five semantic classes obtained from the dataset: terrain, buildings, low points, bridges, and vegetation. Luo et al. [124] proposed a tree detection algorithm through a deep learning framework based on a multi-channel information complementarity illustration. An adapted graph convolution network with local topological information was developed to extract the ground class thus avoiding parameters selection that did not consider different ground topographies. Then, a multichannel representation in addition to Multi-Branch Network (MBNet) was used through fusing multi-channel features. Corte et al. [125] used uncrewed aerial vehicle LiDAR point cloud to test four different ML approaches to detect individual trees and estimate their metrics such as diameter at breast height, total height, and timber volume. The tested methods were SVM, RF, NN, and Extreme Gradient Boosting. Windrim and Bryson [126] isolated individual trees, determine stem points, and further built a segmented model of the main tree stem that encompasses tree height, and diameter. This approach used deep learning models passing through multiple stages starting by ground characterization and removal, delineation of individual trees, and segmentation of tree points into stem and foliage. An example of output of their algorithm is shown in Figure 5. For extracting grasses and individual trees, Man et al. [90] extracted the two-dimensional distribution map of urban vegetation using the object-based RF classification method. Chen et al. [127] employed a PointNet network [27] for segmenting the individual tree crowns using the voxelization strategy.

Finally, Vayghan et al. [3] extracted high-elevation objects from the LiDAR data using the developed scan labelling method, and then the classification methods of a NN. Adaptive Neuro-Fuzzy Inference System (ANFIS), and Genetic Based K-Means algorithm (GBKMs) were used to separate buildings and trees with the purpose of evaluating their performance.

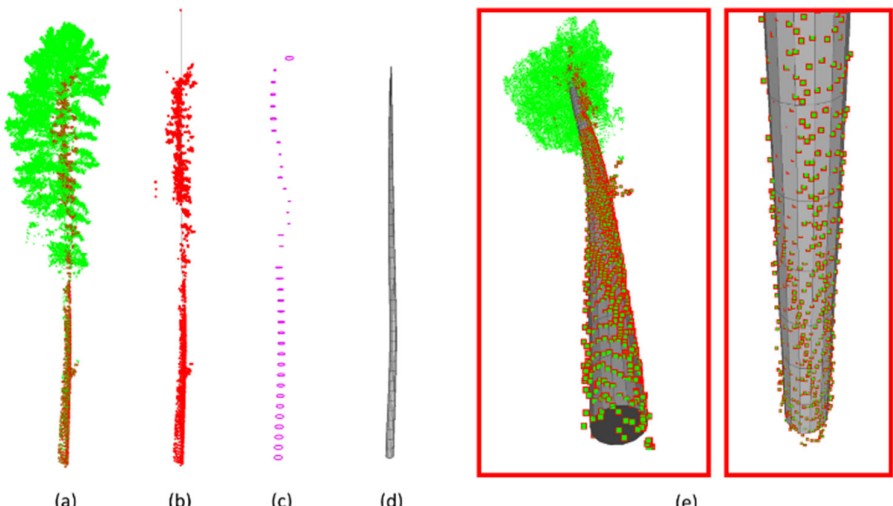

**Figure 5.** An example of the output of Windrim and Bryson's [126] (Figure 8) deep learning model where (**a**) is the segmented point cloud, (**b**) isolated stem points, (**c**) RANdom SAmple Consensus (RANSAC) algorithm circles attach stem section and (**d**) refined stem sections estimate based on robust least-squares fitting process. Panel (**e**) shows examples of the final fitted stem model.

### 5.4. Classification of Tree Species

Zou et al. [26] suggested a voxel-based deep learning method to classify terrestrial LiDAR point clouds of a forested area into species. They used three consecutive steps. After the extraction of individual trees using the density of the point clouds, a low-level feature voxel-based representation was constructed and then the classification of tree species was achieved by using a deep learning model.

Marrs and Ni-Meister [50] compared NNs, k-nearest neighbors, and RF approaches for recognizing tree species. The used variable reduction techniques and showed mixed results depending on the exact set of inputs to each machine learner. Dimensionality reduction based on classification tree nodes is a technique worth trying on multisource datasets. Mizoguchi et al. [128] classified individual tree species using terrestrial LiDAR based on CNN. The key component was the initial step of a depth image creation which well described the characteristics of each species from a point cloud.

Ba et al. [93] employed SVM and RF algorithms to test the discrimination level between tree genera. In this context, tree crowns were isolated and global morphology and internal structure features were computed. Yu et al. [51], Budei et al. [129], and Blomley et al. [57] estimated tree species based on an RF using tree features as predictors and tree species as a response for correctly extracted trees. Figure 6 shows an example of a successful detection phase. Yang et al. [58], and Nguyen et al. [130] both identified the tree species from LiDAR data in addition to other airborne measurements such as hyperspectral images using an SVM classifier. Hell et al. [131] tested the capacity of two deep learning networks PointCNN [132], and 3DmFV-Net [133] for the classification of four different tree species, both living and dead, using LiDAR data. It was shown that 3DmFV-Net is adequate for the geometry of the single trees, whereas PointCNN permits the incorporation of other features.

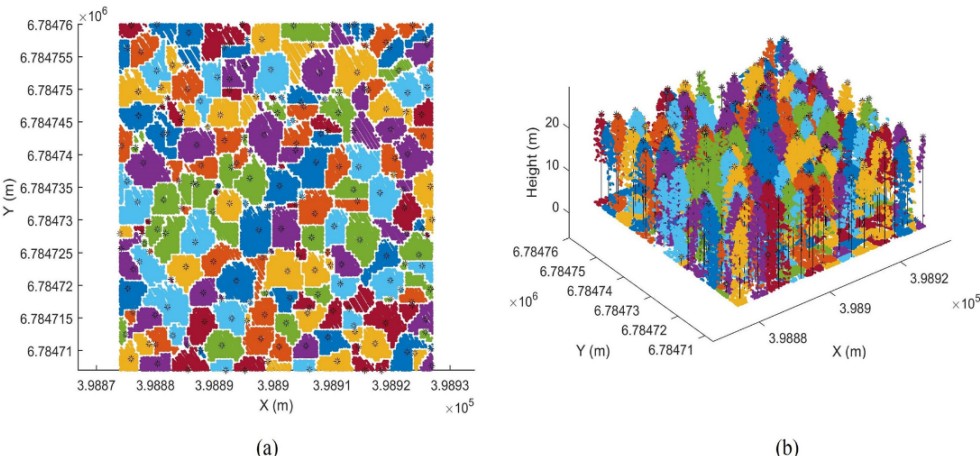

(a)                                                                                         (b)

**Figure 6.** An example of the results of Yu et al. [51] tree detection stage from (**a**) a plan view and (**b**) a 3D view.

### 5.5. Road Marking Classification

The high retro-reflective materials of road markings cause a high laser intensity with respect to the surrounding areas. On one hand, this feature allows easy identification of the road markings but unfortunately, the road markings are not only incomplete but also contain discontinues. That is why the road marking classification represents a challenging task [42]. In this context, Wen et al. [134] used a modified U-net model to segment road marking pixels to overcome the intensity variation, low contrast, and other obstacles. (Ma et al. [135] developed a capsule-based deep learning framework for road marking extraction and classification that consists of three modules. This approach starts with the segmentation of road surfaces. Thereafter, an Inverse Distance Weighting (IDW) interpolation is applied. Based on the convolutional and deconvolutional capsule operations, a U-shaped capsule-based network is created, and a hybrid network is developed using a revised dynamic routing algorithm and Softmax loss function. Fang et al. [42] proposed a graph attention network named GAT_SCNet to simultaneously group the road markings into 11 categories from LiDAR point clouds. The GAT_SCNet model builds serial computable subgraphs and uses a multi-head attention mechanism to encode the geometric and topological links between the node and neighbors to calculate the different descriptors of road marking.

### 5.6. Other Applications

In addition to the main applications presented previously, several important attempts to employ the ML for achieving other automatic operations on LiDAR data are documented in the literature. Ma et al. [136] proposed a workflow for the automatic extraction of road footprints from urban airborne LiDAR point clouds using deep learning PointNet++ [61]. In addition to the point cloud and laser intensity, the co-registered images and generated geometric features are employed to describe a strip-like road. In this context, graph-cut and constrained Triangulation Irregular Networks (TIN) are considered. Shajahan et al. [137] suggested a view-based method called a MultiView Convolutional Neural Network with Self-Attention (MVCNN-SA), which recognizes the roof geometric forms by considering multiple roof point cloud views.

In self-driving cars, several applications such as object recognition, automatic classification, and feature extraction are carried out using ML techniques [34,36,138–142]. The importance of data filtering before starting the modelling operation has been established and Gao et al. [143] proposed a filtering algorithm that uses deep learning and a multi-position sensor comparison approach to eliminate reflection noise. Nurunnabi et al. [144] introduced a local feature-based non-end-to-end deep learning approach that generated a binary classifier for terrain class filtering from which the feature relevance in addition to

the models of different feature combinations were analyzed. Cao and Scaioni [145] applied a deep learning algorithm for semantic segmentation of terrestrial building point clouds. To reduce the number of labels, they suggested a label-efficient deep learning network (3DLEB-Net) that obtained per-point semantic labels of building point clouds with limited supervision. Shokri et al. [100] proposed an SVM approach for automated detection of powerlines from a LiDAR point cloud. In roadside laser scanning system applications, Zhang et al. [146] addressed the goal of a joint detection and tracking scheme by applying PV-RCNN [147] to automatic vehicle and pedestrian detection from the measured moving point cloud. Yin et al. [148] established a squeeze-excite mechanism in local aggregation procedures and employed deep residual learning through a suggested deep learning network that classified complicated piping elements. Amakhchan et al. [149] applied an MLP to filter the LiDAR building point cloud by eliminating the non-roof points. Mammoliti et al. [150] applied the semi-supervised clustering which combined semi-supervised learning and cluster analysis, to evaluate the rock mass discontinuities, orientation and spacing.

## 6. Conclusions

This paper has summarized and reviewed the state-of-the-art ML approaches applied to topographical LiDAR data. Four aspects were considered to analyze the studied methods. First, although all suggested approaches use an airborne or terrestrial LiDAR point cloud of the scanned scene as input data, some of them use, sometimes simultaneously, additional data such as real images, multispectral images, and waveforms to improve their efficiency. Of course, prima facie, using supplementary data may improve the conditions for obtaining the target result, but it is worth considering the contribution of the additional data to the final result. How critical the additional data are to the success of the target task needs to be verified.

Second, in literature, a long list of supervised and unsupervised ML techniques is available. As the unsupervised methods do not need labelled data, the use of these methods can solve the training data labelling problem. Unfortunately, most of the suggested approaches focus only on three supervised ML techniques: NN, RF, and SVM. More research is necessary to investigate the possible application, on LiDAR data, of other ML techniques, especially the unsupervised variety. These may provide opportunities for more efficient and lower cost solutions.

The third aspect is the concept of the LiDAR point cloud structure used within ML algorithms. Many of the proposed algorithms try to transform the question of 3D LiDAR data processing into 2D imagery processing so as to exploit the availability of the image processing informatics tools. These transformations lead to loss of information partly because of dimension reduction. Furthermore, the data reduction through downsampling techniques is similar to the pooling operation employed in image processing algorithms. This procedure is undesirable because it leads to the loss of information which may be beneficial to classify the data successfully. In this context, more research is needed to design a new methodology that simultaneously conserves the LiDAR data and saves the processing time.

Fourth, in regard to the new tools or trends for large-scale mapping and 3D modelling, ML techniques can mainly be employed to achieve five operations on topographical LiDAR data which are: buildings class detection, data classification, point cloud segmentation into vegetation and non-vegetation classes, separation of different tree species, and road marking classification. Some other applications of ML appear rarely in the literature. In fact, most feature-detection operations from topographical LiDAR data can be carried out with the help of classification procedures such as the detection of lines, planes, vertices, surfaces, breaklines, and borders. Filtering operations and modelling also represent an investigation area to apply ML techniques. Clearly, more effort and investigation are needed to improve and to apply ML algorithms on topographical LiDAR data.

**Author Contributions:** Conceptualization: F.T.K., Z.G. and G.C.; Methodology: Z.G. and F.T.K.; Investigation: F.T.K.; Resources: F.T.K. and Z.G.; Writing—original draft preparation: F.T.K., Z.G. and G.C.; Writing—review and editing: F.T.K., Z.G. and G.C.; Visualization: F.T.K. and Z.G.; Supervision: F.T.K., Z.G. and G.C. All authors have read and agreed to the published version of the manuscript.

**Funding:** This research received no external funding.

**Data Availability Statement:** Figure 4 was adopted from Ekhtari et al. [119], Figure 5 was adopted from Windrim and Bryson's [126], and Figure 6 was adopted from Yu et al. [51].

**Acknowledgments:** Thanks to Paul Reed Managing Director of East Coast Surveys (Aust) Pty Ltd. and CloudXPlus company to provide the dataset of Figure 1 which was measured in Queensland, Australia, http://www.eastcoastsurveys.com.au (accessed on 9 August 2022).

**Conflicts of Interest:** The authors declare no conflict of interest.

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
