# Peer review of "Review of Automatic Processing of Topography and Surface Feature Identification LiDAR Data Using Machine Learning Techniques"

_remotesensing, doi:10.3390/rs14194685_

Round 1

Reviewer 1 Report

This article reviews the important and recent studies in the area of Machine Learning (ML) applications on LiDAR data .

The authors explore various domains (in a context where these approaches are flourishing): 

- the input data, coming from LIDAR but also from other data, multispectral image, intensity, color ...  , 

- The use of unsupervised ML to try to solve the problem of labeling training data. 

-The design of the LiDAR point cloud structure used in ML algorithms. Many proposed algorithms attempt to transform the issue of processing 3D LiDAR data into 2D image processing in order to exploit the availability of image processing computational tools. Some of these approaches are mentioned in this paper 

- finally, some applications of ML techniques on LIDAR data are discussed (only the most 'relevant' for the authors)  

- detection of building classes, 

- segmentation of point clouds into vegetation and non-vegetation classes, separation of different tree species, and classification of road markings. 

Of course, the subject is very vast and unless a book is devoted to it, it was right to limit the field of investigation.

However, we may regret a certain 'neutral' form of this work. Indeed, we may reasonably ask ourselves about the usefulness of a review work and this in all review works. 

As far as I am concerned, I would like more implication from the authors. Indeed, they are colleagues with experience in this field and it seems to me that they should be more implicated in the work and not limit themselves, for the most part, to a list of publications accompanied by a few words to describe the method used by the authors they cite.

I would have liked perhaps less quotation but a work which could help me to choose a method: a description of the performances of the methods, for example the authors cite the approaches based on the voxel and everyone naturally wonders about the performances... Likewise, are these approaches implementable? open-source? 

Finally, a reasoned discussion on the validity, the precision of the methods, a discussion led by experts such as the authors, a discussion that would implicate them, would be, it seems to me, more interesting and constructive than a list of publications, most of them summarily described. 

But these remarks are unfortunately applicable to most of the review works and seem to affect the readers little, under these conditions and with the restrictions posed by the authors, this article fits well in the standard of the review articles and thus can be published as is.

Author Response

This article reviews the important and recent studies in the area of Machine Learning (ML) applications on LiDAR data.

The authors explore various domains (in a context where these approaches are flourishing): 

- the input data, coming from LIDAR but also from other data, multispectral image, intensity, color ...  , 

- The use of unsupervised ML to try to solve the problem of labeling training data. 

-The design of the LiDAR point cloud structure used in ML algorithms. Many proposed algorithms attempt to transform the issue of processing 3D LiDAR data into 2D image processing in order to exploit the availability of image processing computational tools. Some of these approaches are mentioned in this paper 

- finally, some applications of ML techniques on LIDAR data are discussed (only the most 'relevant' for the authors)  

- detection of building classes, 

- segmentation of point clouds into vegetation and non-vegetation classes, separation of different tree species, and classification of road markings. 

Of course, the subject is very vast and unless a book is devoted to it, it was right to limit the field of investigation.

However, we may regret a certain 'neutral' form of this work. Indeed, we may reasonably ask ourselves about the usefulness of a review work and this in all review works. 

As far as I am concerned, I would like more implication from the authors. Indeed, they are colleagues with experience in this field and it seems to me that they should be more implicated in the work and not limit themselves, for the most part, to a list of publications accompanied by a few words to describe the method used by the authors they cite.

I would have liked perhaps less quotation but a work which could help me to choose a method: a description of the performances of the methods, for example the authors cite the approaches based on the voxel and everyone naturally wonders about the performances... Likewise, are these approaches implementable? open-source? 

Finally, a reasoned discussion on the validity, the precision of the methods, a discussion led by experts such as the authors, a discussion that would implicate them, would be, it seems to me, more interesting and constructive than a list of publications, most of them summarily described. 

But these remarks are unfortunately applicable to most of the review works and seem to affect the readers little, under these conditions and with the restrictions posed by the authors, this article fits well in the standard of the review articles and thus can be published as is.

Answer:

Thank you very much for spending your time and efforts reviewing our manuscript. We ‎appreciate all your observations and comments. In fact, we agree with your opinion about the ‎limitations of the review papers in the general case. Moreover, you underline the necessity of ‎publishing a book that details this topic and illustrates the testing results of most of the machine ‎learning approaches suggested in the literature. However, publishing a book about this topic ‎represent a gigantic work. Moreover, testing ML approaches may need a considerable time to ‎realize. The research acceleration in this area forces us to follow the strategy of “step by step” ‎which is why this paper represents a step that will be followed by the step chain which achieved ‎the‏ ‏thematic strategy‏ ‏depicted through your comments.   ‎

Reviewer 2 Report

1 In "2 input data", the author introduces the various point cloud types of input ML. We suggest adding a set of images to make the data presentation more straightforward.

2 In "2.1.1 Airborne LiDAR point cloud" on page 3, lines 106-107 describe the problem with the Airborne LiDAR point cloud input machine learning, and lines 108-109 explain that other researchers have found a breakthrough point for this problem. Unfortunately, there is too little description of the solution here, and the core of the method is not clearly presented. We suggest to expand this section.

3 In "4 Employed ML techniques", RF and SVM are introduced in two subsections, which are less used in recent years and both are more basic classification models. We suggest to combine them into one section.

4 In "3 Conceptions of point cloud structure for applying ML algorithms" and "4 Employed ML techniques", the authors have compiled many references, but lack a summary of each subsection. We suggest to summarize the strengths, weaknesses and subsequent developments of the algorithm (framework) at the end of each section.

5 The title "5.2 Data classification" is too general. We suggest that it be modified to be more specific, e.g. "Scene segmentation".

Author Response

Thank you very much for spending your time and efforts for reviewing our manuscript. We note from your feedback that you reviewed the whole paper thoroughly. We appreciate the great efforts that you spent for this paper. We would like to confirm you that the paper has been revised and edited by considering all your comments. All corrections are now highlighted in yellow colour.

Question 2-1.
1 In "2 input data", the author introduces the various point cloud types of input ML. We suggest adding a set of images to make the data presentation more straightforward.

Answer:

Figure 1 has now been provided to address this concern.

Question 2-2.

2 In "2.1.1 Airborne LiDAR point cloud" on page 3, lines 106-107 describe the problem with the Airborne LiDAR point cloud input machine learning, and lines 108-109 explain that other researchers have found a breakthrough point for this problem. Unfortunately, there is too little description of the solution here, and the core of the method is not clearly presented. We suggest to expand this section.

Answer:

Section 2.1.1 is extended as follows:

Airborne LiDAR point clouds provide two obstacles to the applications of ML ‎techniques they are variation in point density within the scanned scene [11] and the large number of LiDAR points [17]. Point density plays a vital role in selecting the neighboring points for ‎the calculation of point features [9]. Point density can vary markedly within the same point cloud with the location within the scanning strip, the terrain topography and the ‎geometry, and the orientation of the scanned object with regard to the scan line all having an affect [8]. For a large area, the data volumes can be excessive, meaning the training step will place heavy demands on the computer capacity and processing time [17; 18]. Lin et al. [19] and Mao et al. [20] developed approaches to mitigate this problem and classify an urban point cloud into nine classes: ‎powerlines, low vegetation, impervious surfaces, cars, fences, roofs, façades, shrubs, and ‎trees. In this context, Mao et al. [20] developed a Receptive Field Fusion-and-Stratification Network ‎‎(RFFS-Net). An innovative Dilated Graph ‎Convolution (DGConv) and its extension, the ‎Annular Dilated Convolution (ADConv), as fundamental components of elementary building ‎blocks. The receptive field fusion procedure was applied ‎with the Dilated and Annular Graph ‎Fusion (DAGFusion) component. Thus, the detection of dilated and annular graphs with ‎numerous receptive zones allows ‎the acquisition of developed multi-receptive field feature ‎implementation‏ ‏to improve classification accuracy.

To efficiently extract only one class from the urban point cloud, Ao et al. [21] advised ‎using a presence and background learning algorithm like a backpropagation neural ‎network.

Question 2-3.

3 In "4 Employed ML techniques", RF and SVM are introduced in two subsections, which are less used in recent years and both are more basic classification models. We suggest to combine them into one section.

Answer:

Sections 4.1and 4.2 are combined into one section as follows:

4.1- Random Forest (RF) and Support Vector Machine (SVM)

Tarsha Kurdi et al. [86] summarized the applications of RF classifiers for automatic vegetation ‎detection and modelling using LiDAR point clouds. Many authors used RF exclusively on ‎LiDAR data [87], whereas other authors used additional data [88, 89]. Yu et al. [90], and Yu et al. [51] estimated tree characteristics such as ‎diameter, height, and stem volume using a RF classifier and Levick et al. [91] connected the DSM and field-measured wood volume using a RF algorithm. Chen et al. [87] ‎used the feature selection method and a RF algorithm for landslide detection under forest canopy, where ‎the DTM and the slope model were constructed for the scanned area, and ‎the features were calculated at the pixel level. The same principle was used by Guan et al. [92] to identify the city classes in urban areas and Ba et al. [93] employed RF for ‎detecting the tree species.

Man et al. [89] applied a RF classifier to calculate a ‎two-dimensional distribution map of urban vegetation. In this study, individual tree ‎segmentation was conducted on a CHM and point cloud data separately ‎to obtain three-dimensional characteristics of urban trees. The results show that both the RF classification and object-based classification could extract urban vegetation accurately, ‎with accuracies above 99%, and the individual tree segmentation based on point cloud data ‎could delineate individual trees in three-dimensional space better than CHM ‎segmentation. Arumäe et al. [94] calculated a model for predicting necessity thinning ‎using the RF technique to retrieve the two indicative parameters for requiring thinning, height percentage and the canopy ‎cover.‎ Park and Guldmann, [63] used a RF algorithm to classify building point ‎clouds into four classes: rooftop, wall, ground, and high outlier. To overcome the complexity ‎of building geometry of the Ming and Qing Dynasties’ Official Architecture style (MQDOAs), Dong ‎et al. [95] employed semantic roof segmentation. This method was composed of two stages. Some ‎geometric features such as the normalized symmetrical distance, relative height, and local height ‎difference are extracted and then the RF algorithm is applied to classify the roof point cloud. ‎Feng and ‎Guo, [64] suggested a segment-based parameter learning approach in which a 2D ‎land cover map is chosen to generate labelled samples, and a ‎formalized operation is then ‎implemented to train the RF classifier. ‏Liao et al. [96] fed in point cloud super voxels and ‎their convex connected patches into a RF algorithm. For this purpose, they consider ‎three types of features: point-based, eigen-based, and grid-based. ‎

The SVM algorithm tries to find a hyperplane in high dimensional feature space to ‎classify some linearly correlative point distributions. While there could be many hyperplanes that separate the target classes, the hyperplane that optimizes the boundary between the classes is identified. Aside from just linear classification, SVM can carry out nonlinear classification using the kernel trick by indirectly drawing their inputs into high-dimensional feature spaces [85].

Though the SVM classifier is efficient for data classification when using rather small data, it is ‎also used by Ba et al. [93] to recognize tree species. ‎Murray et al. [43] trained a SVM on the passing and ongoing results of a CNN ‎algorithm ‎through pixel classification and the interpolation result of the intensity vector as input data.‎ Hoang et al. [97] introduced a hybrid approach of a CNN and a SVM for 3D shape recognition, ‎where eight layers of the CNN are utilized for geometric feature extraction and afterward a SVM is applied ‎to classify them. ‎Zhang et al. [98] suggested an object-based approach to classify an urban airborne LiDAR point ‎cloud. First, different point features such as geometry, radiometry, topology, and echo ‎characteristics are extracted and then the SVM classifier algorithm was applied to detect five classes: ‎terrain, vegetation, building, powerlines, and vehicles.‎ To detect powerlines, Shokri et al. [99] eliminated the undesirable points and then apply ‎the SVM after calculating the point geometric features.‎

In conclusion, RF and SVM are less used in recent years, and both are more basic ‎classification models. Therefore, most modern approaches focus on deep learning techniques.

Question 2-4.

4 In "3 Conceptions of point cloud structure for applying ML algorithms" and "4 Employed ML techniques", the authors have compiled many references, but lack a summary of each subsection. We suggest to summarize the strengths, weaknesses and subsequent developments of the algorithm (framework) at the end of each section.

Answer:

A new paragraph is added to the end of Section 3.1 as follows:

However, voxelization tries to conserve the LiDAR point cloud 3D structure by de-fining a ‎spatial matrixial form that enables managing improved management of the point cloud. Hence, the form will be limited by the available, the used memory and the ‎requested required processing time. may represent the main facing limitations.‎

A new paragraph is added to the end of Section 3.2 as follows:

To conclude, the use of graphic structure facilitates the point cloud processing duty tasks by using image processing functions, but unfortunately at the cost of minimizing, the 3D structure advantages will be minimized. 

A new paragraph is added to the end of Section 3.3 as follows:

In fact, this approach allows applying all operations directly on the point cloud, but it still requires an optimized neighborhood searching procedure.

A new paragraph is added to the end of Section 3.4 as follows:

Though the downsampling reduces the data volume, it loses an important information ‎quantity that ‎may be useful to object recognition and modelling. ‎‎

A new paragraph is added to the end of Section 4.1 as follows:

In conclusion, RF and SVM are less used in recent years, and both are more basic ‎classification models. Therefore, most modern approaches focus on deep learning techniques.

A new paragraph is added to the end of Section 4.3 as follows:

Finally, despite the promising results obtained by deep learning as well as encoder-decoder structure, more focus is needed on unsupervised learning techniques which may cancel the request for training data.

Question 2-5.

5 The title "5.2 Data classification" is too general. We suggest that it be modified to be more specific, e.g. "Scene segmentation".

Answer:

The title "5.2 Data classification” is converted to “Scene segmentation”

Reviewer 3 Report

Τhis is a Review Paper. It reviews Automatic Processing of LiDAR Data Using Machine Learning Techniques. The paper is generally well-written. It can be improved if typical examples found in the bibliography are given with relevant figures and discussion.

Author Response

Thank you very much for spending your time and efforts for reviewing the manuscript. Your feedback does not represent simple comments only, it outlines a roadmap for a high-quality paper. Hence, it is a great pleasure for us to consider all the suggested comments and answering them one by one as follows: 

Question 3-1.

Τhis is a Review Paper. It reviews Automatic Processing of LiDAR Data Using Machine Learning Techniques. The paper is generally well-written. It can be improved if typical examples found in the bibliography are given with relevant figures and discussion.

Answer:

  Figures 1, 4, 5, and 6 are added and discussed as illustrated in the revised manuscript. 

Reviewer 4 Report

The authors of this manuscript provide a description of research work related to the use of machine learning in topographical lidar applications.

I found the manuscript to be well researched, with a comprehensive list of references touching on many aspects of this field.  However, I have several concerns with the manuscript as written.

First, the title, introduction, and conclusion lead the reader to believe that this manuscript serves as a review of machine learning techniques applied to a broad range of lidar applications, when in fact, this is a review of one specific field of study: topography and surface feature identification lidar. This manuscript does not mention the wide variety of machine learning applications occurring outside of topography, e.g., in atmospheric remote sensing of winds, aerosols, and clouds to name a few. The title, introduction, and conclusion need to specify the much narrower scope of this review to avoid misleading the reader. For example, the statement on line 719, “applied to LiDAR data” should read “applied to topographical LiDAR data” and the statement on line 744 that “ML techniques are mainly employed to achieve five operations on LiDAR data” may be true for the specific area of topography described in this manuscript but is not true when applied to all lidar applications as implied in this sentence.

Second, I unfortunately found the manuscript to be difficult to read and digest due to use of extensive jargon from this specific research area with little effort at even basic definition. While serving as an excellent guide to major contributors to the work being done in this field, I believe this manuscript would be mostly opaque to graduate students new to this topic and others who are outside this field, significantly limiting the manuscript’s potential audience and scope. For example, the paragraph starting on line 244 lists nine ML methods but makes no effort to define them or discuss their advantages and disadvantages. Most of the methods are not discussed at all within the rest of the manuscript. I found that the readability also suffers from paragraphs like the one starting on line 570 that forces the reader to read through eight lists of all the slightly different classes that eight different authors used, rather than just stating the conclusion that starts on line 156: classes are chosen depending on the study area and study aim. Is it useful to list this much detail or can the writing be made more concise? Similarly, I thought the manuscript could benefit from helpful figures, such as a diagram of how the authors see the field being laid out, instead of just relying on the section titles to define the authors’ vision. Could figures from some of the referenced papers be included to illustrate key points that the authors are trying to make? 

Finally, I believe the manuscript would benefit from more in-depth discussion of the authors’ perspectives on the field and subsequent conclusions and recommendations for further actions. The final section of the manuscript is titled “Conclusion and perspective” but the “perspectives” are essentially summarized in the last sentences of each of the concluding paragraphs as: criticality of additional data used in concert with the lidar data needs to be validated, and more research is needed in the other areas. I do not find that these statements add new insight to the scientific body of knowledge. This review is a contribution to the special collection entitled “New Tools or Trends for Large-Scale Mapping and 3D Modelling”, however, this manuscript does not clearly delineate what are the new tools in the field relative to the old tools and does not discuss overall trends.

Minor specific comments and edits are included below: 

Line 10: “has widely addressed” should be “has been widely addressed”.

Line 14: Recommend deleting “an” before “ML”.

Line 42: “becomes” should be “has become”.

Line 100-101: “ML techniques they are variation…” should probably be “ML techniques: variation…”

Line 114: What do the authors mean by “a static and mobile terrestrial LiDAR point cloud”?  Do the authors mean static or mobile?

Line 123: Why are the references in this line listed out of order and why are some separated by commas and some by semicolons?

Line 145: Suggest deleting “they”.

Line 171: Here and throughout the manuscript, “LiDAR point cloud” should be “LiDAR point clouds”. Please verify the correct use of plural or singular throughout.

Line 244: There should be a colon after “methods”.

Line 248: Should “Conceptions” be “Concepts”?

Line 300: It is confusing that the section title is “Graphic structure”, yet the first line of this section talks about “graph structure”. Are these the same?

Line 696: Why are these references listed out of order?

Author Response

The authors of this manuscript provide a description of research work related to the use of machine learning in topographical lidar applications.

I found the manuscript to be well researched, with a comprehensive list of references touching on many aspects of this field.  However, I have several concerns with the manuscript as written.

Answer:

Thank you very much for taking the time to review our manuscript. We note from your feedback that you reviewed the paper very carefully line by line. We appreciate the great effort you put into this task. We feel extremely fortunate that our paper has received much attention from you. In fact, such high-quality reviewing reflects the outstanding quality of the Remote Sensing Journal. This is why we thank the journal editor for the careful selection of the referee team. At this stage, we would like to confirm that the paper has been revised and edited by considering all your comments. All corrections are now highlighted in yellow. I hope that the revised version meets the required standard and is accepted for publication in this journal.

Question 4-1.

First, the title, introduction, and conclusion lead the reader to believe that this manuscript serves as a review of machine learning techniques applied to a broad range of lidar applications, when in fact, this is a review of one specific field of study: topography and surface feature identification lidar. This manuscript does not mention the wide variety of machine learning applications occurring outside of topography, e.g., in atmospheric remote sensing of winds, aerosols, and clouds to name a few. The title, introduction, and conclusion need to specify the much narrower scope of this review to avoid misleading the reader. For example, the statement on line 719, “applied to LiDAR data” should read “applied to topographical LiDAR data” and the statement on line 744 that “ML techniques are mainly employed to achieve five operations on LiDAR data” may be true for the specific area of topography described in this manuscript but is not true when applied to all lidar applications as implied in this sentence.

Answer:

The paper title is modified as follows:

Review of Automatic Processing of Topography and Surface Feature Identification LiDAR Data Using Machine Learning Techniques

The abstract, the introduction, and the conclusion are adapted with new paper title. All modifications are highlighted in the corrected manuscript.

Question 4-2.

Second, I unfortunately found the manuscript to be difficult to read and digest due to use of extensive jargon from this specific research area with little effort at even basic definition. While serving as an excellent guide to major contributors to the work being done in this field, I believe this manuscript would be mostly opaque to graduate students new to this topic and others who are outside this field, significantly limiting the manuscript’s potential audience and scope. For example, the paragraph starting on line 244 lists nine ML methods but makes no effort to define them or discuss their advantages and disadvantages. Most of the methods are not discussed at all within the rest of the manuscript. I found that the readability also suffers from paragraphs like the one starting on line 570 that forces the reader to read through eight lists of all the slightly different classes that eight different authors used, rather than just stating the conclusion that starts on line 156: classes are chosen depending on the study area and study aim. Is it useful to list this much detail or can the writing be made more concise? Similarly, I thought the manuscript could benefit from helpful figures, such as a diagram of how the authors see the field being laid out, instead of just relying on the section titles to define the authors’ vision. Could figures from some of the referenced papers be included to illustrate key points that the authors are trying to make? 

Answer:

We thank you so much for offering us this overview discussion about the paper construction. ‎We are so happy to read your words because we agree with you about each word you have ‎cited. You addressed several crucial questions that we discussed before writing the paper. ‎We feel that you were among us in this discussion. Let us take these questions one by one. To ‎whom this paper is addressed? How this paper can be useful? What information the paper can ‎contain and for which information we must send the reader references? What is the paper's ‎role in the current research axis? What is the paper goal?‎

These questions are not all questions, but they represent the more important ones, and their ‎answers must be translated through the paper body. The literature is rich in references to ‎machine learning techniques, we must avoid repeating information explained deeply in ‎literature, so the solution is to send the readers who need details to a suitable reference. This ‎paper is addressed to researchers who focus on the paper topic, which is why the paper tries to stay ‎far from details that could be obtained from references, but simultaneously, we underline the ‎key ideas. This manuscript represents a 4D model of ML applications on LiDAR data. This model ‎enables the user to understand the state of the art of the topic, the link between the suggested ‎approaches, the faced problematics in each branch, the requested solutions, and the useful ‎references, e.g., one researcher desires involve in this research area, this paper saves for him ‎one year of research work. It put on the table all the achieved works in an organized and ‎analysed way which enables him to take suitable decisions. ‎

Concerning the question of line 570, the classification question is widely discussed in this research area. One scanned scene consists of several classes, the question is: do the same classification algorithm can be used to extract the desired class list? Or one algorithm can recognise only certain classes? That is why the decision is taken to identify the classification algorithms according to detected classes, and then the reader can select the algorithm corresponding to his target.

In the context of answering your comments, the next modifications are carried out on the manuscript:

The last paragraph of Section 2.5 is extended as follows:

Finally, Duran et al. [68] compared nine ML methods, logistic regression, linear discriminant analysis, K-NN, decision tree, Gaussian Naïve Bayes, MLP, adaboost, RF, and SVM to classify LiDAR and colored photogrammetric point clouds into four classes: buildings, ground, low and high vegetation with the highest accuracy being attained with MPL. For more details about these ML techniques, please see Mohammed et al. [85] and Kim, [100].

The first paragraph of Section 5.2 is extended as follows:

The classification question is widely discussed in this research area. One scanned scene consists of several classes, and the question that arises is: can the classification algorithm can be used to extract the desired class list? Or can one algorithm only recognize certain classes? For this reason, we have chosen to identify the classification algorithms according to detected classes. With respect to airborne data, not all authors agree about the ideal number of classes. Wen‎ et al. ‎‎[41] developed a deep learning network that classified the airborne LiDAR data into nine classes: ‎powerlines, low vegetation, cars, fences, roofs, facades, shrubs, and trees. Despite Wang and Gu, [118] using the same number of classes, their class list is different: earth bar, grass, ‎roads, buildings, trees, water, powerlines, cars, and ships. Li et al. [76] suggested a deep ‎learning pixel-based analysis network to distinguish four classes in airborne data: trees, grass, ‎roads, and buildings. Another class list is suggested by Ekhtari et al. [119] for airborne data ‎which is: buildings, soil, grass, trees, asphalt, and concrete, whereas Zhao et al. [52] made ‎small modifications to these classes as follows: roads, buildings, grass, trees, powerlines, and soil. ‎Another modification to these classes is suggested by Shinohara et al. [59]: roads, buildings, ‎transmission towers, trees, powerlines, and ground. Liao et al. [96] classified the airborne point ‎cloud into three main classes: terrain, buildings, and vegetation using the RF algorithm. Zhao et al. [120] suggested a Point Expanded Multi-Scale Convolutional Network (PEMCNet) ‎to classify the airborne LiDAR data containing point cloud, intensity, and return number, into five classes: ground, high ‎vegetation, building, water, and raised road. To calculate the point features, it created point expanded grouping units that ‎combined the extracted features at diverse scales.‎ It is fair to say that the classes chosen in each study are a product of the study area and study aim rather than a desire to develop a universal class set.

Figures 1, 4, 5, and 6 are added and discussed as illustrated in the revised manuscript. 

Question 4-3.

Finally, I believe the manuscript would benefit from more in-depth discussion of the authors’ perspectives on the field and subsequent conclusions and recommendations for further actions. The final section of the manuscript is titled “Conclusion and perspective” but the “perspectives” are essentially summarized in the last sentences of each of the concluding paragraphs as: criticality of additional data used in concert with the lidar data needs to be validated, and more research is needed in the other areas. I do not find that these statements add new insight to the scientific body of knowledge. This review is a contribution to the special collection entitled “New Tools or Trends for Large-Scale Mapping and 3D Modelling”, however, this manuscript does not clearly delineate what are the new tools in the field relative to the old tools and does not discuss overall trends.

Answer:

The last paragraph of section 6 is modified as follows:

Fourth, in regard to the new tools or trends for large-scale mapping and 3D modelling, ML techniques can mainly be employed to achieve five operations on topographical LiDAR data which ‎are: buildings class detection, data classification, point cloud segmentation into vegetation and ‎non-vegetation classes, separation of different tree species, and road marking classification. ‎Some other applications of ML appear rarely in the literature. In fact, most feature detection ‎operations from topographical LiDAR data can be carried out with the help of classification procedures such as ‎the detection of lines, planes, vertices, surfaces, breaklines, and borders. Filtering operations ‎and modelling also represent an investigation area to apply ML techniques. Clearly, more effort ‎and investigation are needed to improve and to apply ML algorithms on topographical LiDAR data.

Question 4-4.

Line 10: “has widely addressed” should be “has been widely addressed”.

Answer:

Line 10 is modified as follows:

provided promising results and thus this topic has been widely addressed in the literature ‎during

Question 4-5.

Line 14: Recommend deleting “an” before “ML”.

Answer:

Line 14 is modified as follows:

structure for applying ML, the ML ‎techniques used, and the ‎applications of ML on LiDAR data.

Question 4-6.

Line 42: “becomes” should be “has become”.

Answer:

Line 42 is modified as follows:

use of ML techniques become a popular research topic [5]. ‎

Question 4-7.

Line 100-101: “ML techniques they are variation…” should probably be “ML techniques: variation…”

Answer:

Lines 100-101 are modified as follows:

Airborne LiDAR point clouds provide two obstacles to the applications of ML ‎techniques: variation in point density within the scanned scene [11] and the large number of LiDAR points [17].

Question 4-8.

Line 114: What do the authors mean by “a static and mobile terrestrial LiDAR point cloud”?  Do the authors mean static or mobile?

Answer:

Line 114 is modified as follows:

This subsection focuses only on the ML approaches that use a static or mobile ‎terrestrial LiDAR point cloud as input data either indoors or ‎outdoors.

Question 4-9.

Line 123: Why are the references in this line listed out of order and why are some separated by commas and some by semicolons?

Answer:

Line 123 is modified as follows:

point cloud [17, 27, 28, 29, 30, 31, 32]. Point density variation has less influence ‎in terrestrial

Question 4-10.

Line 145: Suggest deleting “they”.

Answer:

Line 145 is modified as follows:

[39] used an unsupervised domain adaptation ML to classify terrestrial ‎LiDAR data and suggested using Generative Adversarial Network (GAN)

Question 4-11.

Line 171: Here and throughout the manuscript, “LiDAR point cloud” should be “LiDAR point clouds”. Please verify the correct use of plural or singular throughout.

Answer:

Line 171 is modified as follows:

Indeed, ‎numerous authors develop classification ML networks using LiDAR point clouds in ‎addition to digital images as input data. Nahhas et al. [46] employed orthophotos in addition to airborne LiDAR point ‎clouds to recognize the building class by using an ‎autoencoder-based dimensionality reduction to convert low-level features into compressed ‎features.

Question 4-12.

Line 244: There should be a colon after “methods”.

Answer:

Line 244 is modified as follows:

Finally, Duran et al. [68] compared nine ML methods: logistic regression, linear discriminant analysis, K-NN, decision tree, Gaussian Naïve Bayes, MLP, adaboost, RF, and SVM to classify LiDAR and colored photogrammetric point clouds into four classes

Question 4-13.

Line 248: Should “Conceptions” be “Concepts”?

Answer:

Line 248 is modified as follows:

Concepts of point cloud structure for applying ML algorithms

Question 4-14.

Line 300: It is confusing that the section title is “Graphic structure”, yet the first line of this section talks about “graph structure”. Are these the same?

Answer:

Line 300 is modified as follows:

Using graphic structure to transform the 3D point cloud into a 2D regular grid has

Question 4-15.

Line 696: Why are these references listed out of order?

Answer:

Line 696 is modified as follows:

using ML techniques [34, 36, 138, 139, 140, 141, 142].

Round 2

Reviewer 2 Report

I do not have comments in this stage. 

Author Response

Thank you very much for spending your time and efforts reviewing our manuscript. 

Reviewer 4 Report

I thank the authors for their careful consideration of my previous review comments and their diligent responses to my concerns.  I found the manuscript to be significantly improved, especially with the additional figures.  All of my major comments have been addressed. I only have a few minor comments for consideration. After these are addressed, I believe this manuscript will be ready for publication. 

Line 131: The sentence starting as “An innovated Dilated…” seems to be missing a verb. Perhaps it should read “are fundamental” instead of “as fundamental”?

Line 328: Suggest changing “enables managing improved management” to “enables improved management”.

Line 330: This sentence needs attention. There is a period in the middle of the sentence.

Line 370: By “but unfortunately at the cost of minimizing, the 3D structure advantages will be minimized”, do the authors perhaps mean “but unfortunately at the cost of minimizing the 3D structure advantages”?

Line 611: I suggest changing “can be used to extract” to “be used to extract”.

Line 726: Should “from (2) a plan view” be “from (a) a plan view”?

Author Response

I thank the authors for their careful consideration of my previous review comments and their diligent responses to my concerns.  I found the manuscript to be significantly improved, especially with the additional figures.  All of my major comments have been addressed. I only have a few minor comments for consideration. After these are addressed, I believe this manuscript will be ready for publication. 

Answer:

Thank you very much for spending your time and efforts reviewing our manuscript. We ‎appreciate all your observations and comments. We would like to confirm you that the paper has been revised and edited by considering all your comments. All corrections are now highlighted in yellow colour.

Question 4-1.
Line 131: The sentence starting as “An innovated Dilated…” seems to be missing a verb. Perhaps it should read “are fundamental” instead of “as fundamental”?

Answer:

Line 131 is modified as follows:

An innovative Dilated Graph ‎Convolution (DGConv) and its extension, the ‎Annular Dilated Convolution (ADConv), are fundamental components of elementary building ‎blocks

Question 4-2.
Line 328: Suggest changing “enables managing improved management” to “enables improved management”.

Answer:

Line 328 is modified as follows:

spatial matrixial form that enables improved management of the point cloud.

Question 4-3.
Line 330: This sentence needs attention. There is a period in the middle of the sentence.

Answer:

Line 330 is modified as follows:

Hence, the form will be limited by the available, the used memory, and the ‎requested processing time may represent the main limitations.  

Question 4-4.
Line 370: By “but unfortunately at the cost of minimizing, the 3D structure advantages will be minimized”, do the authors perhaps mean “but unfortunately at the cost of minimizing the 3D structure advantages”?

Answer:

Line 370 is modified as follows:

To conclude, the use of graphic structure facilitates the point cloud processing duty tasks by using image processing functions, but unfortunately at the cost of minimizing the 3D structure advantages.

Question 4-5.
Line 611: I suggest changing “can be used to extract” to “be used to extract”.

Answer:

Line 611 is modified as follows:

can the classification algorithm be used to extract the desired class list?

Question 4-6.
Line 726: Should “from (2) a plan view” be “from (a) a plan view”?

Answer:

Line 726 is modified as follows:

tree detection stage from (a) a plan view and (b) a 3D view.
